# Federated High-Dimensional Online Decision Making

**Chi-Hua Wang**                                              *chihuawang@ucla.edu*
*Department of Statistics, University of California, Los Angles*

**Wenjie Li**                                                 *li3549@purdue.edu*
*Department of Statistics, Purdue University*

**Guang Lin**                                                 *guanglin@purdue.edu*
*Department of Mathematics, Statistics, Purdue University*
*School of Mechanical Engineering, Purdue University*

**Reviewed on OpenReview:** *https://openreview.net/forum?id=TjaMO63fc9*

## Abstract

We resolve the main challenge of federated bandit policy design via exploration-exploitation trade-off delineation under data decentralization with a local privacy protection argument. Such a challenge is practical in domain-specific applications and admits another layer of complexity in applications of medical decision-making and web marketing, where high-dimensional decision contexts are sensitive but important to inform decision-making. Existing (low dimensional) federated bandits suffer super-linear theoretical regret upper bound in high-dimensional scenarios and are at risk of client information leakage due to their inability to separate exploration from exploitation. This paper proposes a class of bandit policy design, termed `Fedego Lasso`, to complete the task of federated high-dimensional online decision-making with sub-linear theoretical regret and local client privacy argument. `Fedego Lasso` relies on a novel multi-client teamwork-selfish bandit policy design to perform decentralized collaborative exploration and federated egocentric exploration with logarithmic communication costs. Experiments demonstrate the effectiveness of the proposed algorithms on both synthetic and real-world datasets.

## 1 Introduction

Federated bandits (Huang et al., 2021; Dubey & Pentland, 2020) is an emerging setting of decentralized sequential decision making that emphasizes on collaborate bandit learning and data decentralized decision making. Designing effective decentralized sequential decision making strategies requires resolving the fundamental exploration-exploitation trade-off under *data decentralization with local privacy protection*, that is, clients do not submit their raw data in any circumstance and one client should not have any chance to infer decision rule of another client based on shared information. In principle, clients in collaborate bandit learning stage should be coordinated to explore available actions and gain information to inform future decision making. In decentralized decision making stage, one client's decision should not depend on other client's information to avoid risk of indirect information leakage. Thus, coordinate exploration and privacy protected exploitation are central challenges in federated version of exploration-exploitation trade-off. However, such trade-off admits another layer of complexity in the presence of high-dimensional decision context, especially in applications of medical decision making or web marketing (Bastani & Bayati, 2020). Consequently, federated high-dimensional decision making problems lead challenges on both learning and decision making. On learning, the agent should adopt estimation method to handle high-dimensional context with designing smart sampling scheme; on decision making, the agent should only utilize local clients information to give final decision to meet the purpose of federation. As a result, designing a bandit sampling scheme smartly is the key to minimize the regret while protect local client privacy with data decentralization.

Table 1: Regret guarantee comparison for federated bandit with linear feedback functions. $M$ : number of clients; $K$ : number of arms; $T$ : time horizon; $d$ : dimension of the decision context.

| Bandit algorithms | Regret (High-d $d = O(T)$ ) | Communication cost |
|---|---|---|
| Centralized (Dimakopoulou et al. (2017)) | $O(T\sqrt{KM\log^3(KMT)})$ | $O(Md^2KT)$ |
| Decentralized (Amani & Thrampoulidis (2020)) | $O(T^{3/2}\sqrt{M}\log(MT))$ | $O(dM^2)$ |
| Fed-PE (Huang et al. (2021)) | $O(T\sqrt{KM\log(KMT)})$ | $O\left(Md^2K\log T\right)$ |
| Async-LinUCB (Li & Wang (2021)) | $O(T^{3/2}(\log T)^2)$ | $O(dM\log T)$ |
| Byzantine-Robust (Jadbabaie et al. (2022)) | $O(T^{7/4}M)$ | $O(T^{1/2})$ |
| Lasso Bandit(Bastani & Bayati (2020)) | $O((\log M + \log T)^2 K s_0^2)$ | No communication |
| Teamwork Lasso (Wang & Cheng (2020)) | $O(M(\log T)^2 K s_0^2)$ | $O(KMd\log_2(T/Kq))$ |
| Fedego Lasso(This work) | $O((\log M)(\log T)^2 K s_0^2)$ | $O(KMd\log_2(T/Kq))$ |

Existing works on federated bandits (Huang et al., 2021; Li & Wang, 2021; Jadbabaie et al., 2022), unfortunately, only protect the local privacy of data but not the local privacy of 'information'. That is, one client's decision making, in current arts of federated bandits, are using other client's information. While their method protect local data privacy, they share the resulting information freely to inform collaborate decision making at the risk of indirect information leakage. More frequent the clients communicate with the central server, more risk on indirect information leakage. Such risk of information leakage is from the Upper Confidence Bound (UCB) approach (Auer, 2002; Li et al., 2010) utilized in current federated bandits works. The reason is that the UCB approach cannot handle exploration and exploitation separately. In addition, Table 1 shows that existing federated (low-dimensional) bandit works suffer super-linear regret in the region of high-dimensional scenario, where the decision horizon $T$ scales with the decision context dimension $d$, i.e. $T = O(d)$ Bastani & Bayati (2020).

To avoid super-linear regret in the high-dimensional scenario require rethinking on exploration and exploitation. Our strategy is to design the decision making policy with (i) decentralized collaborative exploration and (ii) federated egocentric exploitation. In (i), we coordinate all clients to commit same decision no matter their current decision context to study that decision's efficacy. Such coordination prevents leakage of decision information and results in decentralized collaborative exploration. In (ii), we allow clients to commit their own reward-maximizing decision to minimize the regret. Such selfish decision protect client decision information and results in federated egocentric exploitation.

**Major Contributions.** We deliver a novel architecture for federated high-dimensional online decision making and associated algorithms. Following that, we highlight our major contributions. (i) **Fedego Lasso Algorithm.** Federated-egocentric Lasso algorithm (Fedego Lasso) exploits Lasso regression to learn sparse local reward models and to inform future decision without compromising local data privacy. In particular, it enables each client to compute a personalized, low-dimensional local reward model, which we refer to as the client private Lasso, that utilizes client's unique decisions, actions and observations of local data. (ii) **Convergence Rate and Regret bound.** We establish convergence results of three type of Lasso estimates implemented in the proposed Fedego Lasso algorithms (Lemma 3.1, 3.4 and 3.6). Such convergence guarantees is not trivial given the non-i.i.d. properties of dataset collected during online decision making process (Remark 1.2). With these convergence rate results, we establish a theoretical regret upper bound (Theorem 3.7) for the algorithm performance of Fedego Lasso. (iii) **Empirical Results.** Through a combination of synthetic and real datasets (PharmGKB, Medical MNIST) we show the benefits of Fedego Lasso in (a) benefits of federation architecture and (b) benefits of recruiting more clients on further regret reduction and faster error rate convergence in real-world tasks including personalize dosage searching and medical image labeling.

**Benefits of Fedego Lasso.** We list benefits of Fedego Lasso over standard linear contextual bandit learning (that learns a single reward model with no central server to share and no high dimensional decision context). (I) *More efficient, effective and secure decision making procedure.* By sharing the online-learned Lasso regression, each client can make more efficient and effective local updates at each communication round. Such update is beneficial in committing its own individual decision making. Besides, our federation architecture reduce the risks of indirect data leakage or inverse decision rule recovery. This is unlike standard

linear contextual bandit learning where in a heterogeneous setting requires several rounds of model updates to recover oracle policy, and thus *hurts* performance. (II) *Provable federation architecture for online decision making with high-dimensional decision context.* Existing works on linear contextual bandit-based online decision making with high-dimensional decision context focus on the effect of sparsity-inspired methods on explore-exploit trade-off. Current state of related literature does not have work designing federation architecture to further facilitate explore-exploit trade-off in bandit learning. To our knowledge, this is the first provable federation architecture for online decision making with high-dimensional decision context that demonstrates the benefit of cooperation.

**3 Key Enhancements in Practical Applicability of `Fedego Lasso` over `Teamwork Lasso` and `Lasso Bandit`.** Compared to `Teamwork Lasso` in Wang & Cheng (2020) and `Lasso Bandit` in Bastani & Bayati (2020), our `Fedego Lasso` bridges the gap between data privacy restrictions and the need for effective decision making in joint data utilization scenarios. The first is to allow data sharing in multi-institutional contexts. While `Teamwork Lasso` assumes "complete data sharing" scenario, which is not the case in reality, `Fedego Lasso` enhances data sharing by allowing clients keeping their data in house and only share the regression coefficients to do joint decision making. The second is to address privacy concerns from federation. The policy in `Lasso Bandit` and `Fedego Lasso` are *infeasible* in healthcare field due to the restriction from HIPAA regulations, which prevents client to share their data directly to the third party(e.g. central server). Our `Fedego Lasso` are HIPAA-compliant, since we only share the regression coefficient but not the original data to the third party. The last enhancement is to enable privacy compliant joint decision making. Both `Lasso Bandit` and `Fedego Lasso` neglects privacy regulation on data usage, leading risk of violating HIPPA law. We design the policy of `Fedego Lasso` to align the modern data sharing standard, making our algorithms with broader applicability in reality.

**Technical advancements on *Lasso Bandit* and *Teamwork Lasso*.** Here we discuss our technical advancements on extending the theoretical analysis in *Lasso Bandit* and *Teamwork Lasso* to secure logarithm expected regret in federated learning case. The first technical contribution is a finer control on the convergence rate of the federated Lasso $\hat{\beta}^\sharp$ ( equation 5), c.f. Lemma 3.4. This relies on redefining the "good event" in the federated learning case to ensure the probability bound on the "bad event" is independent of the number of client $M$. Such independence builds on a right lower bound (c.f. equation 3) on the phase length $q$ such that the aggregated estimate $\hat{\beta}^\sharp$ to have non-asymptotic convergence rate of order $O(1/M)$. Another key technical contribution is a finer control the convergence rate of the egoistic Lasso $\hat{\beta}^\flat$ (equation 6) such that the contributions to regret is of order $O(\log T)$. This relies on finding a right regularization level for the Lasso procedure to make sure the non-asymptotic convergence rate of $\hat{\beta}^\flat$ is of $O(1/M)$ order, c.f. Lemma 3.6. In conclusion, we refine the proof on the non-asymptotic convergence rate on the Lasso procedure in *Lasso Bandit* and *Teamwork Lasso* such that the final regret of *Fedego Lasso* secure logarithmatic regret.

## 1.1 Related works

**Federated bandits.** Recent works have called attention to the topic of federated bandits. Shi et al. (2021) examine efficient client-server communication and coordination methods for federated MAB with personalization, using heterogeneous reward distributions at local clients. Zhu et al. (2021); Li et al. (2020); Dubey & Pentland (2020) discuss how to protect local data privacy in federated bandits via differential privacy. Unfortunately, existing methods to solve federated linear contextual bandit suffer super-linear regret in high-dimensional scenario ($d = O(T)$), since their cumulative regret scales linearly in context dimension $d$ (Table 1). Our work advances these prior efforts by relaxing low dimension reward model assumptions to the case with high-dimensional decision context.

**Lasso bandits.** Bastani & Bayati (2020) first introduced linear contextual bandit with high-dimensional covariate and proposed the Lasso bandit algorithm. Follow-up works such as Wang et al. (2018) and Wang & Cheng (2020) improved the regret bounds and extended it to different problems. In contrast, another line of literature (Kim & Paik, 2019; Hao et al., 2020; Oh et al., 2021; Li et al., 2022) proposed different style of algorithms for solving high dimensional bandit problems. However, these efforts consider a completely different bandit problems where $K$ contexts are observed at each round and only one underlying parameter $\beta$ is present for all arms. One way to possibly transfer results in Bastani & Bayati (2020) and its follow ups into the other line of research, and compare them, is to concatenate all the parameter $\beta_k$'s for different

arms into one long vector $\widetilde{\beta}$ (see notations in Section 1.2), and assume that the $K$ contexts generated in each round are always "one-hot" with the same feature for all the arms. Our works advances the setting in Bastani & Bayati (2020) by designing federation architecture to ensure local data privacy security while delineating explore-exploit tradeoff in online decision making with high-dimenstional decision context.

**Comparison with `Teamwork Lasso Bandit` in Wang & Cheng (2020).** Our bandit sampling policy design (Figure 1) and client-center communication scheme (Figure 2) advances the "alternating two stage" design in Wang & Cheng (2020) in the regard of client privacy protection. Previous work Wang & Cheng (2020) on batch decision making also solves the federated decision making problem in this paper, by setting their `Teamwork Lasso Bandit` with a batch of clients. However, such approach requires all clients share their raw data to server to train a central Lasso to inform decision, while our `Fedego Lasso` allows clients keep their data, only share their individual Lasso estimates, and use aggregated Lasso to inform decision. **Such fundamental difference of our `Fedego Lasso` advances `Teamwork Lasso Bandit` in Wang & Cheng (2020) into a more private exploration and exploitation design.** For exploration stage (Blue; Figure 1), while Wang & Cheng (2020) access all clients' raw data for a teamwork Lasso estimate, our `Fedego Lasso` do not touch client's raw data but only their individual Lasso estimate for a federated Lasso estimate (Federation step at Figure 2). For exploitation stage (Red; Figure 1), while Wang & Cheng (2020) access all clients' raw data for a selfish Lasso estimate, our `Fedego Lasso` allows clients to do egocentric decision by keeping their local estimate private and do not share to center or other clients. While Wang & Cheng (2020) raise algorithmic privacy concerns in the scope of federated decision making, our work makes contributions to federated high-dimensional decision making problems.

## 1.2 Notations and basic problem formulation

Here we give problem formulation and define the bandit algorithms regret and the Lasso estimator.

**Federated high-dimensional online decision making problems.** We investigate a federated linear contextual bandits scenario in which $M$ *clients* are pulling the same set of $K$ arms, represented by $[K] := 1, 2, \ldots, K$. Each client $m \in [M]$ at each time $t \in [T]$ pulls an arm $k \in [K]$ based on historical information. The expected reward of pulling arm $k$ at decision context $x$ is the inner product $\langle x, \beta_k \rangle$ between $x$ and the reward parameter $\beta_k$. Additionally, there is a sparsity parameter $s_0 \in [d]$, defined as the smallest integer such that for all $k \in [K]$, $\|\beta_k\|_0 \leq s_0$.

**Regret of bandit algorithms $\pi$.** A bandit algorithm $\pi$ of client $m$ pulls arm $\pi_t^{(m)}$ at decision step $t$. The objective is to design bandit algorithms $\pi$ that minimizes the expected cumulative regret among all clients, defined as: $\text{Regret}(T) = \mathbb{E}\left[\sum_{m=1}^{M}\sum_{t=1}^{T}\left(\left\langle x_t^{(m)}, \beta_{k_t^{(m),*}} - \beta_{\pi_t^{(m)}}\right\rangle\right)\right]$, where $k_t^{(m),*} \in [K]$ is a context-specific optimal arm such that : $\forall l \neq k_t^{(m),*}, \langle x_t^{(m)}, \beta_{k_t^{(m),*}}\rangle \geq \langle x_t^{(m)}, \beta_l\rangle$.

In this work, we approach the above federated high-dimensional online decision making problems with LASSO regressions.

**Definition 1.1.** (Lasso regression estimator) Given a dataset $\mathcal{D} = \{(X, Y)\}$, where $Y$ is a $|\mathcal{D}|$-dimension response vector and $X$ is a $|\mathcal{D}| \times d$ decision context matrix from the dataset $\mathcal{D}$. The *Lasso regression estimator* with regularization level $\lambda \geq 0$ is defined as

$$\hat{\beta}\left(\mathcal{D}, \lambda\right) \equiv \arg\min_{\beta}\left\{|Y - X\beta\|_2^2/|\mathcal{D}| + \lambda\|\beta\|_1\right\}. \tag{1}$$

*Remark* 1.2. (**Non-i.i.d. properties of dataset $\mathcal{D}$.**) Due to the nature of online decision making, the Lasso estimate is trained on datasets $\mathcal{D}$ that cannot satisfy typical distributional (or i.i.d.) assumptions in the literature of federated learning. The key reason is that the decision at certain time steps are depending on previous history, leading to dependency between decision and historical data. Therefore, standard convergence theory is not applicable for analyzing Lasso trained on the dataset $\mathcal{D}$. Fortunately, it is still possible to analyze and design the statistical properties of the trained Lasso estimate *if* we carefully design the bandit policy to coordinate the exploration across clients and exploitation within clients during whole online decision making process. See Figure 1 and Section 3.1 for bandit sampling design and Section 3.2 for formal convergence results of trained Lasso estimates.

*Remark* 1.3. (One the usefulness of Lasso regression estimate) We use Lasso since our application is in e-commerce and medical setting. In such setting, the context vector is often high-dimensional, but only

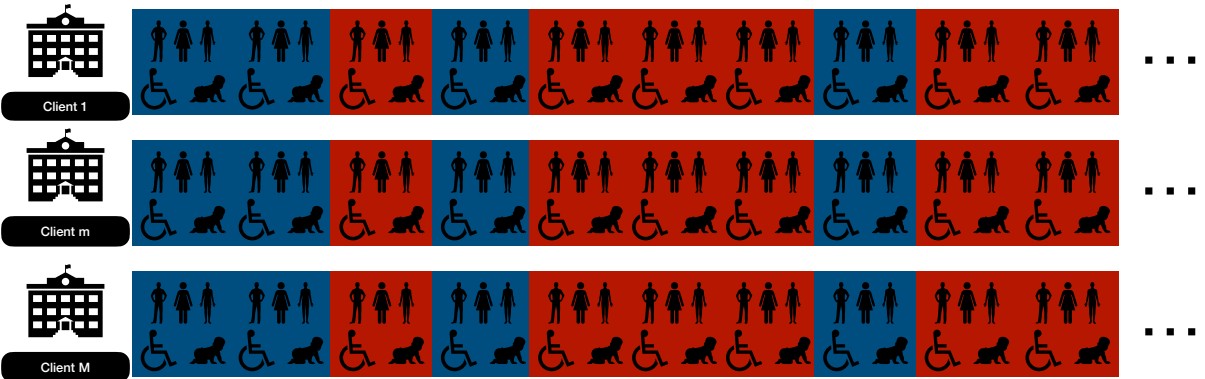

Figure 1: Centralized teamwork-selfish sampling policy. In the teamwork stage (Blue), all clients pull prescribed arms, no matter the current decision context, to form collaborate exploration for a quick recovery of arm reward parameters. In the selfish stage (Red), all clients pull the arm with highest estimated reward for current decision context to form egocentric exploitation for maximizing their own cumulative reward. Each stage has length $Kq$, where $K$ is number of arm and $q$ is the teamwork unit length (See Section 3.1 for details.)

few number of variable are decisive. Lasso captures such property by assuming the sparse structure of the regression coefficients. With such sparsity structure of regression coefficient, we leverage the technique in high-dimensional statistic in our theory (Theorem 3.7). The `Lasso bandit` Bastani & Bayati (2020) paper explains that in medical applications, we might want to identify a small number of biomarkers that are associated with a certain disease. In e-commerce applications, we might want to select a small number of features that are relevant for predicting user behavior. Both of these problems can be formulated as a sparse linear regression problem, where we want to minimize the number of non-zero coefficients in the regression model. Lasso is particularly useful in these cases because it performs both feature selection and regularization simultaneously. By penalizing the L1 norm of the regression coefficients, Lasso encourages sparsity and tends to produce models with a small number of non-zero coefficients. This can lead to better generalization performance and easier interpretability of the resulting model. Therefore, the motivation for using Lasso is to take advantage of its ability to perform feature selection and regularization simultaneously, which is particularly useful for applications where sparsity is assumed.

## 2 Federated online decision making

This section elaborates a horizontal federated learning (as defined in Yang et al. (2019)) version of bandit problems with high-dimensional decision context and linear arm rewards (as introduced by Bastani & Bayati (2020) and others). The three main challenges arise from federated contextual bandits are *(A)* Model local dataset heterogeneity. *(B)* Tighten local data and decision rule privacy security. *(C)* Coordinate efficient central communication. Section 2.1 established our framework to resolve these challenges.

### 2.1 Resolution to federated high-dimensional bandit challenges

**A. Client local bandit models for local dataset heterogeneity.** To account for the intrinsic correlation between rewards associated with different clients pulling the same arm, we assume the *local bandit model* of client $m$ that the expected reward of pulling arm $k$ is a linear function of the decision context $x_t$; formally,

$$r(x_t) \equiv \langle \beta_k, x_t \rangle + \epsilon_t^{(m)}. \tag{2}$$

The parameter $\beta_k \in \mathbb{R}^d$ is a constant but unknown vector for each arm $k \in [K]$. The noise process $\{(\epsilon_t^{(m)}, \mathcal{F}_t^{(m)})\}_{t \in [T]}$ is assumed to be a $\sigma-$subGaussian martingale difference sequence (that is, $E[\epsilon_t^{(m)} \mid \mathcal{F}_{t-1}^{(m)}] = 0$ and $\mathbb{E}[\exp(\lambda \epsilon_t^{(m)}) | \mathcal{F}_{t-1}^{(m)}] \leq \exp(\sigma^2 \lambda^2 / 2)$ for all real $\lambda$). See Section 2.2 for essential statistical regularity assumptions for analyzing federated contextual bandit algorithms. The *local dataset heterogeneity* is depicted naturally by the local bandit model equation 2, since clients have their own decision context sequence $\{x_t^{(m)}\}_{t=1}^T$, resulting heterogeneous reward distributions.

**B. Federation with decision rule hiding to tighten local privacy security.** Our proposal to resolve local data privacy dilemma is via Teamwork-Selfish sampling policy (Figure 1). By *local data privacy*, we

meant that one client should not have any chance to infer decision rule of another client based on shared information. To protect local data privacy, we design federation architecture and hide clients' decision rules from central server. Our approach, in the spirit of the doubling trick (Besson & Kaufmann, 2018), do not require the agents to share to the server their datasets, no matter the datasetes are collected from Teamwork mode or Selfish mode. The only information to share is the Lasso estimates trained on datasets collected during Teamwork mode at certain pre-specified decision points. Thus, the Teamwork-Selfish bandit policy resolves the local data privacy security challenge.

**C. Horizontal federation to coordinate efficient communication.** In horizontal federated-learning system, $M$ clients with same online decision making task collaboratively learn a model with the help of server. We resolve explore-exploit dilemma via client-center communication protocol (Figure 2), established at Section 3.3. In our scheme, each client trains two Lasso estimates, one from Teamwork dataset (for exploration) and the other from Teamwork-Selfish aggregation dataset (for exploitation). Our strategy is to introduce two different modes for agent: Teamwork mode and Selfish mode. Clients only upload their Lasso estimates at the end of the Teamwork mode, thus the communication cost is $\log(T/Kq)$. Our approach, which in spirit is an applications of horizontal federated learning architecture (Yang et al., 2019), coordinates efficient communication between clients and center to perform near-optimal decision making and achieve logarithmic regret performance. Consequently, the communication protocol (Figure 2) resolves the local data privacy security challenge.

## 2.2 Statistical Regularity Conditions

In section 2.2, we list essential statistical regularity assumptions towards convergence rate analysis of Lasso regression estimator (Definition 1.1) and regret analysis of algorithms. We note that these assumptions are standard in the literature of high-dimensional decision making (Bastani & Bayati, 2020).

**Assumption 2.1.** For all clients $m \in [M]$, there exists $C_0 \geq 0$ such that for arms $k_1 \neq k_2$ in $\mathbb{A}$, the distribution of decision context $X$ satisfy $\mathbb{P}(|\langle X, \beta_{k_1}^{(m)} - \beta_{k_2}^{(m)} \rangle| \in (0, \kappa)) \leq C_0 \kappa$ for given $\kappa > 0$.

Assumption 2.1 is known as *Margin Condition* in the classification literature Tsybakov et al. (2004). Such condition ensures only a minor fraction of decision context is sampled near the classification boundary $\{x : \langle x, \beta_{k_1} - \beta_{k_2} \rangle = 0\}$ in which efficacy of both arms are indistinguishable (Rusmevichientong & Tsitsiklis, 2010; Wang & Cheng, 2020).

**Assumption 2.2.** For all clients $m \in [M]$, there exists a optimality gap constant $h$ and two mutually exclusive arm subsets $\mathbb{A}_{opt}$ and $\mathbb{A}_{sub}$ with $[K] = \mathbb{A}_{opt} \cup \mathbb{A}_{sub}$ such that (a) For each arm $k$ in $\mathbb{A}_{sub}$, it holds for every decision context $x \in \mathcal{X}$ that $\langle \beta_k, x \rangle < \max_{a \in \mathbb{A} \setminus \{k\}} \langle \beta_a, x \rangle - h$ and (b) For each arm $k$ in $\mathbb{A}_{opt}$, there exists a constant $p_* > 0$ of minimal sampling probability such that $\min_{k \in \mathbb{A}_{opt}} \mathbb{P}(X \in U_k) \geq M p_*$ where $U_k \equiv \{x \in \mathcal{X} | \langle \beta_k, x \rangle > \max_{a \in \mathbf{A} \setminus \{k\}} \langle \beta_a, x \rangle - h\}$.

Assumption 2.2 is known as the *Arm Optimality Condition* (Bastani & Bayati, 2020; Wang & Cheng, 2020). Such condition separates arms into an optimal subset $\mathbb{A}_{opt}$ and a suboptimal subset $\mathbb{A}_{sub}$, in the sense that (a) all sub-optimal arms are strictly sub-optimal for every decision context and (b) each optimal arm $k \in \mathbf{A}_{opt}$ is strictly optimal for some decision context ( $U_k$ at Assumption 2.2).

**Assumption 2.3.** For a client $m$, there exists a constant $\phi_0 > 0$ such that for each optimal arm $k \in \mathbf{A}_{opt}$, its population covariance matrix $\Sigma_k \equiv E[XX^\top | X \in U_k]$ belongs to the compatibility set with respect to the true parameter $\beta_k$. That is, $\Sigma_k \in \mathcal{C}(\text{supp}(\beta_k), \phi_0)$, where
$$\mathcal{C}(I, \phi) \equiv \left\{ A \in \mathbb{R}_{\succeq 0}^{p \times p} \mid \forall v \in \mathbb{R}^p \; \|v_I\|_1^2 \leq |I|(v^\top A v)/\phi^2 \text{ if } \|v_{I^c}\|_1 \leq 3 \|v_I\|_1 \right\}.$$

Assumption 2.3 is referred to as the *Compatibility Condition* in high-dimensional statistics (Bühlmann & Van De Geer, 2011). It ensures that the Lasso regression estimator trained on samples $X \in U_k$ converges to the true parameter $\beta_k$ with high probability as the number of samples grows to infinity.

## 3 `Fedego Lasso` algorithms

Algorithm 1 presents our solutions, the `Fedego Lasso` bandit algorithms, to the federated online decision making problems established at Section 2. There are 3 key components in our design of `Fedego Lasso` algorithms: the teamwork-selfish bandit sampling strategy (Figure 1), the clients-central server communication

protocol (Figure 2) and local data privacy preserving. Section 3.1 establishes the teamwork-selfish bandit sampling strategy to implement sparsity-award collaborate exploration. Section 3.2 illustrates how `Fedego Lasso` protect local clients privacy. Section 3.3 establishes the communication protocol of resulting Lasso estimates between clients and central server towards optimal regret performance of online decision making task.

---

**Algorithm 1** `Fedego Lasso`: client $m$

---

**Require:** Decision horizon $T$, number of arms $K$, optimality gap $h^{(m)}$, Teamwork stage $\mathbb{T}$.
  **for** $t = 1 \cdots T$ **do**
    Observe the decision context $x_t^{(m)}$
    **if** $t \in \mathbb{T}$(Teamwork mode) **then**
      $\pi(x_t^{(m)}) = \texttt{ColExplore}(x_t^{(m)}, t)$ (Alg 2)
    **end if**
    **if** $t \notin \mathbb{T}$ (Selfish mode) **then**
      $\hat{K} = \texttt{FedScreen}(x_t^{(m)}, \{\hat{\beta}_{k,t}^{\sharp}\}_{k=1}^{K}, h^{(m)})$ (Alg 3; where $\hat{\beta}_{k,t}^{\sharp}$ is the central federated Lasso estimate defined at equation 5)
      $\pi(x_t^{(m)}) = \texttt{EgoCommit}(\hat{K}, \{\hat{\beta}_{k,t}^{\flat,(m)}\}_{k=1}^{K})$ (Alg 4; where $\{\hat{\beta}_{k,t}^{\flat,(m)}$ is the private egoistic Lasso estimate defined at equation 6)
    **end if**
    Pull the arm $\pi(x_t^{(m)})$, receive reward $r_t^{(m)}$.
  **end for**
  **Return**: Cumulative reward $R(T) = \sum_{t=1}^{T} r_t^{(m)}$.

---

### 3.1 Teamwork-Selfish bandit sampling

This section presents the three key elements implementing the teamwork-selfish bandit sampling strategy (Figure 1): sampling mode, resulting datasets and their Lasso estimates.

**Sampling modes: `Teamwork` and `Selfish`.** Every local client agent has two sampling modes: `Teamwork` mode (Blue block in Figure 1) and `Selfish` mode (Red block in Figure 1). In the `Teamwork` mode, all clients runs *collaborate exploration*, aiming at a quick recover of the arm reward parameter set $\{\beta_k\}_{k=1}^{K}$. In `Selfish` mode, clients run *egocentric exploitation* individually, aiming at an optimal decision for their own current decision context. The design of alternating sampling guarantees the convergence of Lasso while optimizing algorithm performance.

**Datasets: teamwork set $\mathcal{T}$ and selfish set $\mathcal{S}$.** Every local client agent maintains two datasets: teamwork dataset $\mathcal{T}$ and selfish dataset $\mathcal{S}$. In general, datasets $\mathcal{T}^{(m)}$ and $\mathcal{S}^{(m)}$ collect samples from Teamwork and Selfish mode respective for the client $m$. In particular, notations $\mathcal{T}_{[t],k}^{(m)}$ and $\mathcal{S}_{[t],k}^{(m)}$ denote the teamwork and selfish dataset respectively from pulling arm $k$ during decision period $[t] = \{1, 2, \cdots, t\}$. Such maintenance is to separate the data source. Technically, the teamwork set's decision is public due to agreement of all clients. The selfish set's decision is private and only accessible by the client itself. Theoretically, the teamwork set's samples are independently distributed since in which the decisions are independent of the previous history, while in selfish set the samples are dependent since the decisions are dependent on the history.

**Estimates: teamwork Lasso $\hat{\beta}^{\sharp}$ and private Lasso $\hat{\beta}^{\flat}$** Every local client agent maintains two Lasso estimates: client teamwork Lasso and local private Lasso. In principle, the client teamwork Lasso $\hat{\beta}^{\sharp} = \hat{\beta}(\mathcal{T})$ is from running Lasso regression (Definition 1.1) on the Teamwork dataset; in contrast, local private Lasso $\hat{\beta}^{\flat} = \hat{\beta}(\mathcal{T} \cup \mathcal{S})$ is trained on the aggregation dataset of Teamwork and Selfish set. Such maintenance is to to separate federation and decision making. In `Fedego Lasso` algorithm, all local clients upload their client teamwork Lasso to central server at the end of each Teamwork mode to facilitate federation. At each decision step in Selfish mode, all clients commit final decisions based on local private Lasso estimate.

### 3.2 Decision rules in Teamwork and Selfish mode

This section presents the three key subroutines implemented in the `Fedego Lasso` bandit algorithms (Algorithm 1): collaborate exploration (Algorithm 2), federated screening (Algorithm 3) and egocentric commitment (Algorithm 4).

### A. `ColExplore`: Collaborate exploration

Here we introduce 3 key quantities in the Collaborate exploration stage in Algorithm 1. The phase length $q$ determines the block size of each stage in Figure 1, reflecting the communication frequency. The Teamwork stage $\mathbb{T}$ is the teamwork stage (blue block) in Figure 1, in which the decision in this stage only depends on the timestamp but not depends on previous response history. Last, the sample collected at the teamwork stage is used to learn the Lasso regression coefficient for each client and to be shared to the central server.

---

**Algorithm 2** `ColExplore`$(x, t)$

---

   Input: decision context $x$, decision step $t$
   Output: $\pi(x) \equiv (t \mod K)$

---

***A.(1)****Phase length $q$.* A key technical challenge is to determine the phase length $q$ to ensure Lasso estimator convergence is fast enough to informal optimal decision making.

$$q \geq q_{\text{teamwork}} \equiv \max\{\frac{20}{Mp_*}, \frac{4C_2(\phi_1)^2}{Mp_*}, \frac{512x_{\max}^2 \log d}{C_1(\phi)M^2p_*^2h^2}\}(4 + \log M) \tag{3}$$

***A.(2)*** *Planning of Teamwork stage $\mathbb{T}$.* Each block in Figure 1 contains $Kq$ decision steps. The planning of teamwork stage is $\mathbb{T} = \cup_{n=1}^{\lfloor \log_2(T/Kq) \rfloor} \mathbb{T}_n$, where the *nth* Teamwork stage is

$$\mathbb{T}_n = [(2^n - 1)Kq + 1 : 2^n Kq].$$

***A.(3)*** *Client teamwork Lasso.* The client teamwork Lasso for client $m$ at time step $t$ is defined by running Lasso regression (Definition 1.1) in the teamwork dataset $\mathcal{T}_{[t],k}^{(m)}$; formally,

$$\hat{\beta}_{k,t}^{\sharp,(m)} \equiv \hat{\beta}(\mathcal{T}_{[t],k}^{(m)}, \lambda_1). \tag{4}$$

Lemma 3.1 justifies the convergence of client teamwork Lasso equation 4.

**Lemma 3.1.** *(Deviation inequality of client teamwork Lasso) For all arms $k \in [K]$, the client teamwork Lasso estimate equation 4 satisfies*

$$\mathbb{P}\left(\|\hat{\beta}_{k,t}^{\sharp,(m)} - \beta_k\|_1 > h/4x_{\max}\right) \leq 5/Mt^4$$

*if $\lambda_1^{(m)} = \phi_0^2 Mp_*h/64s_0x_{\max}$ and $q \geq q_{teamwork}$.*

*Proof.* See Section A. Lemma 3.1 generalizes the forced sampled Lasso in Bastani & Bayati (2020) into horizontal federated learning scheme. The key difference is to redefine the length of exploration $q_{\text{teamwork}}$ to ensure faster concentration. Finding $q_{\text{teamwork}}$ is non-trivial due to the centralized exploration scheme. Our found solution is to inflate the length of exploration by a $\log M$ factor, that is, $q_{\text{teamwork}} = O(q_0 \log M)$, where $q_0$ is the exploration length in Bastani & Bayati (2020).

*Remark 3.2.* (On the concern of the situation that a proportion of clients explore the same arms) In current design of Collaborate exploration scheme, a subset of customers may keep exploring the same arms. Such possibility raise concerns on fairness of the decision making procedure. In particular, it is challenging to ensure fairness in the allocation of experimental resources. These customers may miss the opportunity to try their best arm, bearing the experimental costs for joint research. To address this issue, a significant challenge we anticipate is to "define a regret formulation that sufficiently reflects decision fairness." Are there minority groups among the clients? How can we ensure that the results of joint decision-making benefit everyone instead of some clients bearing the costs while others enjoy the outcomes? These are non-trivial questions and challenges we expect to encounter.

**B. `FedScreen`: Federated screening.**

At each time step in Selfish mode, the client screen available arms with central federated Lasso. In this stage, the goal of Algorithm 3 is to screen out sub-optimal arms for a given decision context. The screening is based on the central federated Lasso regression estimate equation 5, whose statistical accuracy is supported by Lemma 3.4. After the screening, we are sure that the remaining candidate arms are located in the $\mathbb{A}_{\mathrm{opt}}$ in Assumption 2.2 and contribute at most $h$ in the formal regret.

---

**Algorithm 3** `FedScreen`$(x, \{\hat{\beta}_k\}_{k=1}^K, h)$

---

Input: decision context $x$, federated estimates $\{\hat{\beta}_k\}_{k=1}^K$ (central federated Lasso estimate equation 5), optimality gap $h$

Output: the candidate set $\hat{K}(x) \equiv \{k \in \mathbb{A} : \langle x, \hat{\beta}_k \rangle \geq \max_{l \in \mathbb{A}} \langle x, \hat{\beta}_l \rangle - h/2\}$

---

*Central federated Lasso.* After receiving all clients' teamwork Lasso estimates, the central server performs federation by computing the central federated Lasso. The central federated Lasso for arm $k$ at decision step $t$ is defined as the average of all client Teamwork Lasso estimates; formally

$$\hat{\beta}_{k,t}^{\sharp} \equiv \frac{1}{M} \sum_{m=1}^{M} \hat{\beta}_{k,t}^{\sharp,(m)}. \tag{5}$$

*Remark* 3.3. Note that the central federated Lasso is the average of teamwork Lasso estimates, which are trained on i.i.d. samples, and therefore has convergence guarantees. Such analytical advantage is due to our policy design that all agents explore the efficacy of pre-specified arms during Teamwork mode (Blue block in Figure 1).

**Lemma 3.4.** *For all arms $k \in [K]$, if $t \geq (Kq)^2$, the central federated Lasso estimate equation 5 satisfies*

$$\mathbb{P}\left( \|\hat{\beta}_{k,t}^{\sharp} - \beta_k\|_1 > h^{(m)}/4x_{\max} \right) \leq 5/t^4.$$

*Proof.* See Section B. The key is due to our tighter control of client teamwork Lasso (Lemma 3.1). Such control ensure the deviation probability of equation 5 is independent of client number $M$.

*Remark* 3.5. One can directly apply force estimator deviation inequality in Bastani & Bayati (2020), but the resulting deviation probability will depends on client number $M$ due to union bound. Our technical contribution is to remove such dependency on $M$ by inflating the exploration length by $\log M$ factor.

**C. `EgoCommit`: Egocentric commitment**

Given the resulted set of optimal arm candidates $\hat{K}(x_t^{(m)})$ for decision context $x_t^{(m)}$ from the federated screening (Algorithm 3), the client agent $m$ commits to the arm with the highest estimated expected reward estimated by the private egocentric Lasso equation 6.

---

**Algorithm 4** `EgoCommit`$(\hat{K}(x), \{\hat{\beta}_k\}_{k=1}^K)$

---

Input: candidate set $\hat{K}(x)$, decision context $x$, private estimates $\{\hat{\beta}_k\}_{k=1}^K$ (private egoistic Lasso estimate equation 6)

Output: $\pi(x) \equiv \arg\max_{k \in \hat{K}(x)} \langle x, \hat{\beta}_k \rangle$

---

*Private egoistic Lasso.* The private egoistic Lasso for client $m$ at time step $t$ is defined by running Lasso regression (Definition 1.1) in the teamwork-selfish aggregate dataset $\mathcal{T}_{[t],k}^{(m)} \cup \mathcal{S}_{[t],k}^{(m)}$; formally,

$$\hat{\beta}_{k,t}^{\flat,(m)} \equiv \hat{\beta}\left( \mathcal{T}_{[t],k}^{(m)} \cup \mathcal{S}_{[t],k}, \lambda_{2,t}^{(m)} \right). \tag{6}$$

Lemma 3.6 justifies the convergence of private egoistic Lasso equation 6.

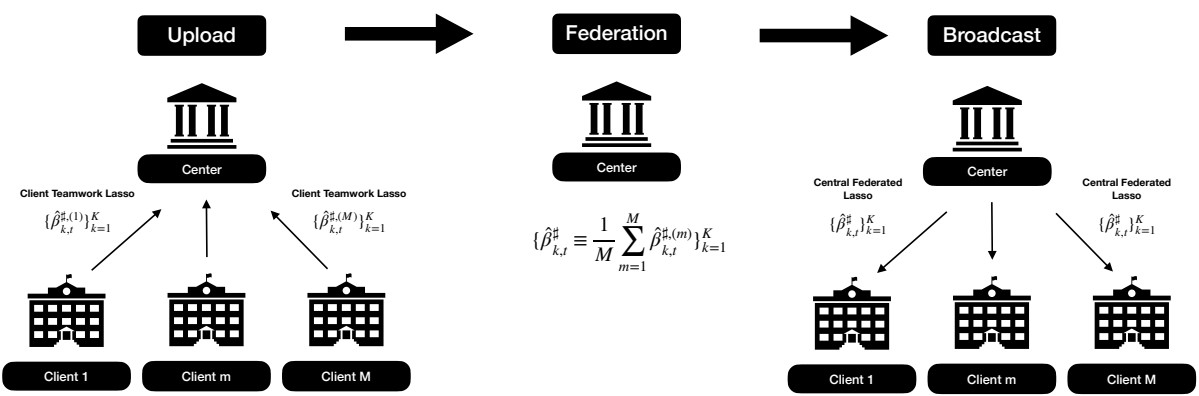

Figure 2: Federated client-center communication. At the end of each Teamwork stage (Blue block in Figure 1), all clients uploads their current teamwork Lasso estimate (defined at equation 4) to the central server. Then, the central server perform federation to compute the federated Lasso estimates (defined at equation 5). Last, the central server broadcast the federated Lasso estimates to all clients for their upcoming federated screening procedure in the Selfish stage (Red block in Figure 1) .

**Lemma 3.6.** *For all optimal arms* $k \in \mathbb{A}_{opt}$, *set* $\lambda_{2,t}^{(m)} = \frac{\phi_0^2}{32 s_0} \sqrt{\frac{M \log dMt}{C_1(\phi_0) p_* t}}$ *and* $t \geq C_5$. *Then, the private egocentric Lasso estimate equation 6 satisfies*

$$\mathbb{P}(\|\hat{\beta}_{k,t}^{\flat,(m)} - \beta_k\|_1 > 16\sqrt{(\log dMt)/(p_*^3 C_1(\phi_0) Mt)})$$
$$\leq 2((Mt)^{-1} + \exp(-p_*^2 C_2^2 M^2 t/32)).$$

*Proof.* See Section C. The key is to refine the estimation error of Lasso estimator. □

### 3.3 Clients-Central Server Communication

Figure 2 presents the three key events in the clients-central server communication protocol: upload, federation and broadcast. In the following we discuss the contribution of such horizontal federated learning architecture to bandit learning and local data privacy security.

**Upload: Clients upload local teamwork Lasso to server.** At the end of each round of the `Teamwork` mode, all clients upload their current client Teamwork Lasso estimates equation 4 to the central server to facilitate the federation. The local teamwork Lasso have difference convergence level due to local data heterogeneity. The collaborate exploration between clients in `Teamwork` mode ensuring decisions of a client will not compromise to the other client. Such advantage is due to design of Teamwork-Selfish sampling strategy, all clients do experiment for collaborate exploration during `Teamwork` mode.

**Federation: Server compute central federated Lasso.** The central sever do averaging to mitigate statistical error of received teamwork Lasso estimates. The outcome estimate is called central federated Lasso (defined at equation 5). The federation event helps security by ensuring internal potential privacy leakage from the central server since clients do not upload any dataset but only their local teamwork Lasso estimates.

**Broadcast: Server broadcast federated Lasso to clients.** After federation, the central server broadcasts to clients the central federated Lasso equation 5, helping client to do valid screening procedure to exclude sub-optimal arms and find out the candidates of optimal arms. Such screening procedure is termed *federated screening* and elaborated at Algorithm 3. The broadcast event helps security by the fact that one client cannot infer other clients' decisions based on the federated Lasso, since the final decision is based on private egocentric Lasso (Eq. 6) as in Algorithm 4. Such a feature helps us avoid internal potential privacy leakage from one client to the other.

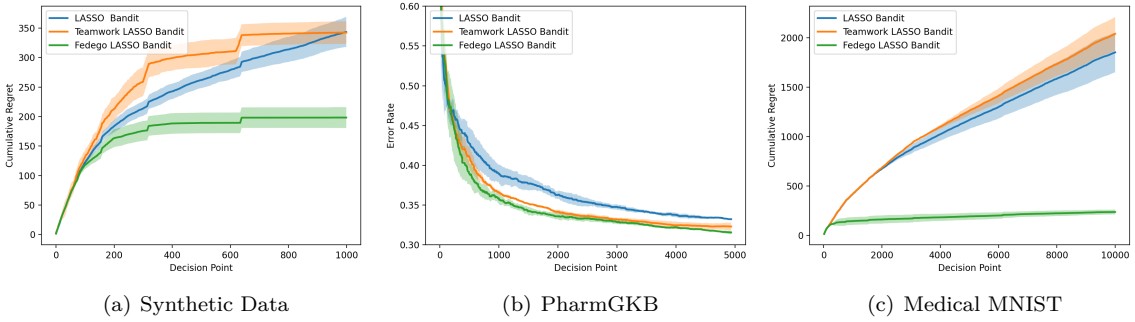

(a) Synthetic Data      (b) PharmGKB      (c) Medical MNIST

Figure 3: Comparison of the performances of `Lasso bandit` in Bastani & Bayati (2020), `Teamwork Lasso` bandit, and `Fedego Lasso` bandit algorithms. Fig (a). Average cumulative regrets per client on synthetic data. Fig (b). Error rates on the PharmGKB dataset. Fig(c). Average cumulative regrets per client on the Medical MNIST dataset.

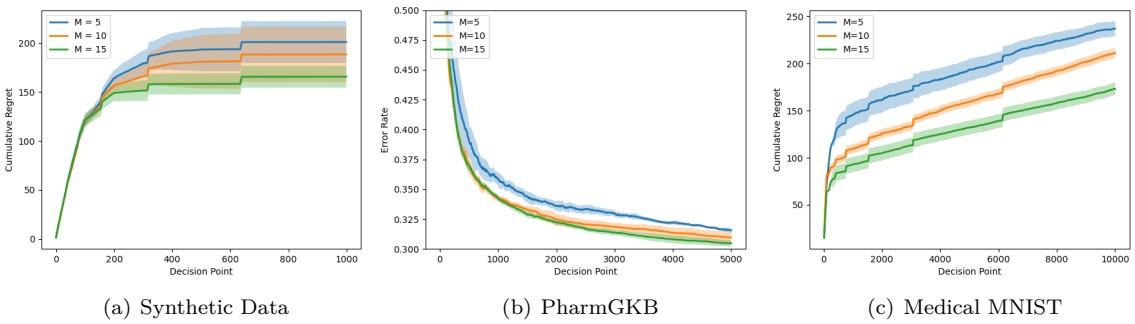

(a) Synthetic Data      (b) PharmGKB      (c) Medical MNIST

Figure 4: The performances when the number of clients increase. Fig (a). Average cumulative regrets per client on synthetic data. Fig (b). Error rates on the PharmGKB dataset. Fig(c). Average cumulative regrets per client on the Medical MNIST dataset.

### 3.4 Communication cost and Regret guarantee of `Fedego Lasso`

The following theorem bounds the regret of `Fedego Lasso` bandit algorithm (Algorithm 1).

**Theorem 3.7.** *The communication cost of `Fedego Lasso` is $O(\log_2(T/Kq))$. The regret of `Fedego Lasso` bandit algorithms $\pi$ over decision horizon $[T]$ satisfies*

$$\boldsymbol{Regret}_\pi(T) = O(\log M[\log T + \log d]^2)$$

*Proof.* See Section D for details. The result supports that `Fedego Lasso` is the first federated bandit algorithms (compared to others in Table 1) that secures sub-linear regret in high-dimensional scenario. Additionally, if we transfer the problem into the setting of Kim & Paik (2019); Hao et al. (2020); Li et al. (2022) as mentioned in Section 1.1, both regret bounds in their works and our work are logarithmic in the dimension $d$. However, their regret bounds are either of order $O(\sqrt{T})$ or $O(T^{2/3})$, while our regret bound is logarithmic in $T$ and the dimension $d$. The advantage comes from not only our unique algorithm design, but also the difference in problem setting and the assumptions in Section 2.2

## 4 Empirical results

While the theoretical regret analysis (Theorem 3.7) provides worst-case guarantees for `Fedego Lasso`, we now examine their performance on a variety of tasks empirically. Examination is conducted using both synthetic and real-world data.

**A. Experiment setup.** We compare `Fedego Lasso` bandit with the vanilla `Lasso` bandit algorithm in (Bastani & Bayati, 2020) and the `Teamwork Lasso` bandit algorithm Wang & Cheng (2020) on the three

datasets. We run each algorithm for 10 independent trials on each dataset and plot the average results with two standard deviations. Additional experimental details are provided in Appendix F. *(a).Synthetic Data.* The sparse parameters $\{\beta_k^{(m)}\}_{k=1}^K$ for each client is generated from randomly sampled support, and then the nonzero parameter values are generated from the uniform distribution on [0,1]. The decision context are drawn randomly from a multivariate normal distribution. *(b).Real data: PharmGKB.* The first real-world task is personalized dosage searching. The Pharmacogenomics Knowledge Base (PharmGKB) dataset was used by Bastani & Bayati (2020) to support the superiority of `Lasso` bandit over the other bandit algorithms. We inherit their settings and investigate the error rates of the two algorithms when giving warfarin dosages based on patient-level decision context such as demographics, diagnosis, and medications. *(c).Real data: Medical MNIST.* The second real-world task is to collaborate classification of Medical MNIST images. To extract the useful features vectors from the medical images, we first train a fully-connected neural network on the dataset till it reaches good training and testing accuracies. At each round of the bandit problem, an image is sampled from the dataset and fed into the neural network. The covariates for the bandit algorithms are the output of an intermediate layer of the neural network. We define the instantaneous regret by whether the image is correctly classified.

**B. Benefits of federation architecture.** Figure 3 supports the benefits of federation architecture enjoyed by the proposed `Fedego Lasso` over the other baselines. In synthetic data scenario, `Fedego Lasso` enjoys substantial regret reduction in Figure 3.(a). In the task of personalized dosage searching, `Fedego Lasso` enjoys faster error rate convergence in Figure 3.(b). In the task of medical image labeling, `Fedego Lasso` again enjoys substantial regret reduction in Figure 3.(c). These empirical evidences supports the benefits of the proposed federation architecture.

**C. Benefits of large number of clients.** Figure 4 supports the benefit of having large number of clients in federated bandit learning. In the synthetic data and the task of medical image labeling, `Fedego Lasso` enjoys further regret reduction by recruiting more clients in bandit learning, as supported by Figure 4.(a) and (c). In the task of personalized dosage searching, `Fedego Lasso` with more clients enjoy faster error rate convergence, as in Figure 4.(b).

## 5 Conclusion

We build a novel architecture on federated linear contextual bandits model with high-dimensional decision context. Such architecture delivers a unified federated bandit framework that resolves local dataset heterogeneity, local data and decision rule privacy security and explore-exploit tradeoff simultaneously. The associated algorithm `Fedego Lasso` utilizes the sparse structure of local bandit models to recover the global parameters and inform efficient federated exploration with effective egocentric exploitation. Theoretical analysis supports that `Fedego Lasso` secures sub-linear regret in the order of $O(\log M(\log T)^2)$ with a communication cost in the order of $O(\log_2(T/Kq))$.

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

# A    Proof of Lemma 3.1

The proof strategy is to use Lemma E.1 on the dataset $(\mathcal{T}^{(m)}_{[t],k})$. We first prove three lemmas for bounding probability of bad events. Then we combines these results to gain deviation inequality of Teamwork Lasso under `FedEgo` bandit sampling scheme.

The following lemma gives a high-probability lower bound of the optimal allocation rate in the dataset $(\mathcal{T}^{(m)}_{[t],k})$.

**Lemma A.1.** *If* $t \geq (Kq)^2$, *the following holds*

$$\mathbb{P}\left(\frac{|(\mathcal{T}^{(m)}_{[t],k})'|}{|(\mathcal{T}^{(m)}_{[t],k})|} > \frac{Mp_*}{2}\right) \geq 1 - \frac{2}{Mt^4}. \tag{7}$$

*Proof.* Recall a version of the Chernoff inequality: given $y$ a sum of mutually independent indicator random variables with expectation $\mu = \mathbb{E}[y]$, one has that $\mathbb{P}[|y - \mu| > \mu/2] < 2\exp(-\mu/10)$.

**Probability upper bound on the expected optimal allocation number.** Consider the indicator of optimal allocation variable $\mathbb{I}(a \in (\mathcal{T}^{(m)}_{[t],k})')$ for all $a \in (\mathcal{T}^{(m)}_{[t],k})$. Assumption 2.2 implies that the expected optimal allocation number $\mu = \mathbb{E}[\sum_{a \in (\mathcal{T}^{(m)}_{[t],k})}]\mathbb{I}(a \in (\mathcal{T}^{(m)}_{[t],k})')$ is at least $Mp_*|(\mathcal{T}^{(m)}_{[t],k})|$; that is, $\mu \geq Mp_*|(\mathcal{T}^{(m)}_{[t],k})|$. In this case, the Chernoff inequality indicate a high-probability upper bound on the optimal allocation number $|(\mathcal{T}^{(m)}_{[t],k})'|$ as

$$\mathbb{P}[|(\mathcal{T}^{(m)}_{[t],k})'| < \frac{Mp_*}{2}|(\mathcal{T}^{(m)}_{[t],k})|] < 2\exp(-\frac{Mp_*}{10}|(\mathcal{T}^{(m)}_{[t],k})|). \tag{8a}$$

Recall that $|(\mathcal{T}^{(m)}_{[t],k})| \geq (1/2)q\log t$. The equation equation 8a becomes

$$\mathbb{P}[|(\mathcal{T}^{(m)}_{[t],k})'| < \frac{Mp_*}{2}|(\mathcal{T}^{(m)}_{[t],k})|] < 2\exp(-\frac{Mp_*}{20}q\log t). \tag{8b}$$

Require $q \geq \frac{20}{Mp_*}(4 + \log M)$. The equation equation 8b becomes

$$\mathbb{P}[|(\mathcal{T}^{(m)}_{[t],k})'| < \frac{Mp_*}{2}|(\mathcal{T}^{(m)}_{[t],k})|] < 2\exp(-(4\log t + \log M)) = \frac{2}{Mt^4}. \tag{8c}$$

$\square$

The following lemma gives the probability upper bound of the failure of compatability condition of sample covariance matrix on dataset $(\mathcal{T}^{(m)}_{[t],k})'$.

**Lemma A.2.**

$$\mathbb{P}\left[\hat{\Sigma}((\mathcal{T}^{(m)}_{[t],k})') \in \mathcal{C}\left(\text{supp}(\beta), \frac{\phi_1}{\sqrt{2}}\right)\right] \geq 1 - \frac{1}{Mt^4} \tag{9}$$

*Proof.* By an application of Lemma EC.6 in Bastani & Bayati (2020), we first have

$$\mathbb{P}\left[\hat{\Sigma}((\mathcal{T}^{(m)}_{[t],k})') \in \mathcal{C}\left(\text{supp}(\beta), \frac{\phi_1}{\sqrt{2}}\right)\right] \geq 1 - \exp(-C_2(\phi_1)^2|(\mathcal{T}^{(m)}_{[t],k})'|) \tag{10a}$$

Recall that $|(\mathcal{T}^{(m)}_{[t],k})| \geq (1/2)q\log t$. Lemma A.1 indicates that with probability at least $1 = 2/Mt^4$,

$$|(\mathcal{T}^{(m)}_{[t],k})'| \geq \frac{Mp_*}{2}|(\mathcal{T}^{(m)}_{[t],k})| \geq \frac{Mp_*}{4}q\log t \tag{10b}$$

Require $q \geq \frac{4C_2(\phi_1)^2}{Mp_*}(4 + \log M)$. The equation equation 10a becomes

$$\mathbb{P}\left[\hat{\Sigma}((\mathcal{T}^{(m)}_{[t],k})') \notin \mathcal{C}\left(\text{supp}(\beta), \frac{\phi_1}{\sqrt{2}}\right)\right] \leq \exp(-(4\log t + \log M)) = \frac{1}{Mt^4}. \tag{10c}$$

$\square$

**Lemma A.3.** *Set* $\lambda_1^{(m)} \equiv \phi_0^2 Mp_* h/64s_0 x_{\max}$. *Then we have,*

$$\mathbb{P}\left(\max_{r \in [d]}\left(2\left|\varepsilon^\top X^{(r)}\right|/|(\mathcal{T}^{(m)}_{[t],k})|\right) \leq \lambda_1^{(m)}/2\right) \geq 1 - \frac{2}{Mt^4} \tag{11}$$

*Proof.* Recall that $|(\mathcal{T}^{(m)}_{[t],k})| \geq (1/2)q\log t$. Require $q \geq \frac{512x_{\max}^2 \log d}{C_1(\phi)M^2 p_*^2 h^2}(4 + \log M)$. we have

$$\exp\left(-C_1(\phi\frac{\sqrt{Mp_*}}{2})|(\mathcal{T}^{(m)}_{[t],k})|(\frac{h}{4x_{\max}})^2 + \log d\right)$$

$$\leq \exp\left(-C_1(\phi)\frac{M^2 p_*^2}{16} \cdot \frac{1}{2}q\log t \cdot \frac{h^2}{16x_{\max}^2} + \log d\right)$$

$$= \exp\left(-\frac{C_1(\phi)M^2 p_*^2 h^2}{512x_{\max}^2}q\log t + \log d\right)$$

$$\leq \exp\left(-(4\log t + \log M)\right) = \frac{1}{Mt^4}$$

$\square$

**Proposition A.4.** *Given*

$$q \geq \max\{\frac{20}{Mp_*}, \frac{4C_2(\phi_1)^2}{Mp_*}, \frac{512x_{\max}^2 \log d}{C_1(\phi)M^2 p_*^2 h^2}\}(4 + \log M).$$

*We have*

$$\mathbb{P}\left(\|\hat{\beta}((\mathcal{T}^{(m)}_{[t],k}), \lambda_1) - \beta_k\|_1 > \frac{h}{4x_{\max}}\right) \leq \frac{5}{Mt}. \tag{13}$$

*Proof.* Combine the above three lemmas.

$$\mathbb{P}\left(\|\hat{\beta}((\mathcal{T}^{(m)}_{[t],k}), \lambda_1) - \beta_k\|_1 > \frac{h}{4x_{\max}}\right)$$

$$\leq \mathbb{P}\left(\max_{r \in [d]}\frac{1}{|(\mathcal{T}^{(m)}_{[t],k})|}|\epsilon^\top X^{(r)}| \geq \lambda_0(\chi, \frac{\phi\sqrt{p}}{2})\right)$$

$$+ \mathbb{P}\left(\hat{\Sigma}(((\mathcal{T}^{(m)}_{[t],k}))') \notin \mathcal{C}(\text{supp}(\beta), \frac{\phi}{\sqrt{2}})\right) \tag{14a}$$

$$+ \mathbb{P}\left(\frac{|((\mathcal{T}^{(m)}_{[t],k}))'|}{|(\mathcal{T}^{(m)}_{[t],k})|} \leq \frac{p}{2}\right)$$

$$\leq \frac{2}{Mt} + \frac{1}{Mt} + \frac{2}{Mt} = \frac{5}{Mt}$$

$\square$

# B   Proof of Lemma 3.4

Recall the definition of central federated Lasso equation 5 $\hat{\beta}_{k,t}^{\sharp} \equiv \frac{1}{M} \sum_{m=1}^{M} \hat{\beta}_{k,t}^{\sharp,(m)}$.

**Step 1** Set the size of error $\chi > 0$.

$$\mathbb{P}\left(\left\|\hat{\beta}_{k,t}^{\#} - \beta_k\right\|_1 > \chi\right) \tag{15}$$

$$=\mathbb{P}\left(\left\|\frac{1}{M}\sum_{m=1}^{M}\hat{\beta}\left(\mathcal{T}_{[t],k}^{(m)},\lambda_1^{(m)}\right) - \frac{1}{M}\sum_{m=1}^{M}\beta_k\right\|_1 > \chi\right) \tag{16}$$

$$=\mathbb{P}\left(\left\|\frac{1}{M}\sum_{m=1}^{M}[\hat{\beta}\left(\mathcal{T}_{[t],k}^{(m)},\lambda_1^{(m)}\right) - \beta_k]\right\|_1 > \chi\right) \tag{17}$$

$$\leq\mathbb{P}\left(\frac{1}{M}\sum_{m=1}^{M}\left\|\hat{\beta}\left(\mathcal{T}_{[t],k}^{(m)},\lambda_1^{(m)}\right) - \beta_k\right\|_1 > \chi\right) \tag{18}$$

$$=\mathbb{P}\left(\sum_{m=1}^{M}\left\|\hat{\beta}\left(\mathcal{T}_{[t],k}^{(m)},\lambda_1^{(m)}\right) - \beta_k\right\|_1 > M\chi\right) \tag{19}$$

$$\leq\mathbb{P}\left(\exists m \in [M] : \left\|\hat{\beta}\left(\mathcal{T}_{[t],k}^{(m)},\lambda_1^{(m)}\right) - \beta_k\right\|_1 > \chi\right) \tag{20}$$

$$\leq\sum_{m=1}^{M}\mathbb{P}\left(\left\|\hat{\beta}\left(\mathcal{T}_{[t],k}^{(m)},\lambda_1^{(m)}\right) - \beta_k\right\|_1 > \chi\right) \tag{21}$$

The first inequality is due to the fact that the $\ell_1$-norm is convex. An application of Jensen's inequality shows that $\|\frac{1}{M}\sum_{m=1}^{M}(\hat{\beta}\left(\mathcal{T}_{[t],k}^{(m)},\lambda_1^{(m)}\right) - \beta_0^{(m)})\|_1 = \frac{1}{M}\|\sum_{m=1}^{M}(\hat{\beta}\left(\mathcal{T}_{[t],k}^{(m)},\lambda_1^{(m)}\right) - \beta_0^{(m)}\|_1 \leq \frac{1}{M}\sum_{m=1}^{M}\|\hat{\beta}\left(\mathcal{T}_{[t],k}^{(m)},\lambda_1^{(m)}\right) - \beta_0^{(m)}\|_1$. The second inequality is due to the fact that the $\ell_1$-norm is non-negative and an application of Pigeonhole principle. The third inequality is an application of Boole's inequality, also known as the union bound.

**Step 2** The client $m$ sets the size of error $\chi = \frac{h^{(m)}}{4x_{\max}}$,

$$\mathbb{P}\left(\left\|\hat{\beta}_{k,t}^{\#} - \beta_k\right\|_1 > \frac{h^{(m)}}{4x_{\max}}\right) \tag{22}$$

$$\leq\mathbb{P}\left(\left\|\hat{\beta}_{k,t}^{\#} - \beta_k\right\|_1 > \frac{\min_{m\in[M]}h^{(m)}}{4x_{\max}}\right) \tag{23}$$

$$\leq\sum_{m=1}^{M}\mathbb{P}\left(\left\|\hat{\beta}\left(\mathcal{T}_{[t],k}^{(m)},\lambda_1^{(m)}\right) - \beta_k\right\|_1 > \frac{\min_{m\in[M]}h^{(m)}}{4x_{\max}}\right) \tag{24}$$

$$\leq M\frac{5}{Mt^4} = \frac{5}{t^4} \tag{25}$$

The first inequality is due to the fact that tail probability is a decreasing function. The second inequality is due to Step 1 above. The third inequality is due to is by the deviation inequality of client Teamwork Lasso estimate (lemma 3.1) by setting the regularization level to be $\lambda_1^{(m)} = (\phi_0)^2 p_* \min_{m\in[M]} h^{(m)}/(64s_0 x_{\max})$.

**Step 3.** Define a "good event"

$$E_t^{(m)} \equiv \cap_{k\in\mathbf{A}}\left\{\left\|\hat{\beta}_{k,t}^{\sharp} - \beta_k\right\|_1 \leq \frac{h^{(m)}}{4x_{\max}}\right\}. \tag{26}$$

The event marks that each central federated Lasso is sufficiently accurate to exclude out sub-optimal arms.

$$\mathbb{P}\left((E_t^{(m)})^c\right) = \mathbb{P}\left(\cup_{k\in A}\left\{\left\|\hat{\beta}_{k,t}^{\#} - \beta_k\right\|_1 \le \frac{h^{(m)}}{4x_{\max}}\right\}\right) \tag{27}$$

$$\le \sum_{k=1}^K \mathbb{P}\left(\left\|\hat{\beta}_{k,t}^{\#} - \beta_k\right\|_1 \le \frac{h^{(m)}}{4x_{\max}}\right) \tag{28}$$

$$\le \frac{5K}{t^4} \tag{29}$$

The first inequality is an application of Boole's inequality, also known as the union bound. The second inequality is due to Step 2 above.

## C  Proof of Lemma 3.6

The proof strategy is to use Lemma E.1 on the dataset $\mathcal{S}_{[t],k}^{(m)}$.

**Lemma C.1.**

$$\mathbb{P}\left(|(\mathcal{S}_{[t],k}^{(m)})'| \ge \frac{Mp_*}{4}t\right) \ge 1 - \exp(-\frac{M^2 p_*^2}{36}t). \tag{30}$$

*Proof.* We have

$$\mathbb{E}[M_{k,t}] \ge \mathbb{P}[A_{n_t} \cap A_{n_t+1}] \cdot \mathbb{P}(X_t \in U_k) \cdot |V_t|$$

$$\ge \frac{8}{9} \cdot Mp_* \cdot \frac{3}{8}t = \frac{Mp_*}{3}t$$

Thus, Hoeffding inequality indicates that

$$\mathbb{P}(M_{k,t} < \mathbb{E}[M_{k,t}] - \eta) \le \exp(-\frac{2\eta^2}{|V_t|}) \le \exp(-\frac{4\eta^2}{t}).$$

In particular, for $\eta = Mtp_*/12$, we have

$$\mathbb{P}(M_{k,t} < \frac{Mp_*}{4}t) \le \exp(-\frac{M^2 p_*^2}{36}t)$$

Thus, the result follows from the fact that $M_{k,t} \le |(\mathcal{S}_{[t],k}^{(m)})'|$.

$\square$

**Lemma C.2.** *For* $t > C_5 \equiv \min_t\{\log Mt < \frac{tM^2 p_*^2 C_2^2(\phi)}{16}\}$

$$\mathbb{P}\left[\hat{\Sigma}((\mathcal{S}_{[t],k}^{(m)})') \in \mathcal{C}\left(supp(\beta), \frac{\sqrt{Mp_*}}{2\sqrt{2}}\phi_1\right)\right] \ge 1 - \frac{1}{Mt} \tag{32}$$

*Proof.* In this case, we have sample size at most $t$ and minimal sampling rate $Mp_*/2$. The first is due to $|\mathcal{S}_{[t],k}^{(m)}| \le t$ and the second is due to $|(\mathcal{S}_{[t],k}^{(m)})'| \ge Mp_*t/4$. Thus, we have $|(\mathcal{S}_{[t],k}^{(m)})'|/|\mathcal{S}_{[t],k}^{(m)}| \ge Mp_*/4$. By definition of $C_5 \equiv \min_t\{\log Mt < \frac{tM^2 p_*^2 C_2^2(\phi)}{16}\}$, we have the probability bound

$$\mathbb{P}\left[\hat{\Sigma}((\mathcal{S}_{[t],k}^{(m)})') \notin \mathcal{C}\left(supp(\beta), \frac{\sqrt{Mp_*}}{2\sqrt{2}}\phi_1\right)\right] \le \exp(-\frac{tM^2 p_*^2 C_2^2}{16})$$

$$\le \exp(-\log(Mt)) = \frac{1}{Mt} \tag{33a}$$

$\square$

**Lemma C.3.** *Set* $\lambda_{2,t}^{(m)} = \frac{\phi_0^2}{32 s_0} \sqrt{\frac{M \log dMt}{C_1(\phi_0) p_* t}}$, *then we have*

$$\mathbb{P}\left( \max_{r \in [d]} \left( 2 \left| \varepsilon^\top X^{(r)} \right| / |\mathcal{S}_{[t],k}^{(m)}| \right) \leq \lambda_{2,t}^{(m)}/2 \right) \geq 1 - \frac{2}{Mt} \tag{34}$$

*Proof.* In this case, we have minimal sampling rate $Mp_*/2$ since $|\mathcal{S}_{[t],k}^{(m)}| \leq t$ and $|(\mathcal{S}_{[t],k}^{(m)})'| \geq Mp_* t/4$. Let $\chi = 16\sqrt{\frac{\log dMt}{p_*^3 C_1(\phi_0) Mt}}$.

$$\exp\left( -C_1(\phi \frac{\sqrt{Mp_*}}{2}) |\mathcal{S}_{[t],k}^{(m)}| \chi^2 + \log d \right)$$
$$\leq \exp\left( -C_1(\phi) \frac{M^2 p_*^2}{16} \cdot \frac{Mp_*}{4} t \cdot 256 \frac{\log(dMt)}{p_*^3 C_1(\phi_0) Mt} + \log d \right)$$
$$= \exp\left( -4M^2 \log(dMt) + \log d \right)$$
$$\leq \exp\left( -(\log t + \log M) \right) = \frac{1}{Mt}$$

$\square$

**Lemma C.4.** *Given $t > C_5$. We have*

$$\mathbb{P}\left( \|\hat{\beta}(\mathcal{S}_{[t],k}^{(m)}, \lambda_{2,t}) - \beta_k\|_1 > 16\sqrt{\frac{\log t + \log M + \log d}{p_*^3 C_1(\phi_0) Mt}} \right) \leq \frac{2}{Mt} + 2\exp\left[ -\frac{p_*^2 C_2(\phi_0)^2}{32} \cdot M^2 t \right]. \tag{36}$$

*Proof.* Apply the above three lemmas

$$\mathbb{P}\left( \|\hat{\beta}(\mathcal{S}_{[t],k}^{(m)}, \lambda_1) - \beta_k\|_1 > \frac{h}{4x_{\max}} \right)$$
$$\leq \mathbb{P}\left( \max_{r \in [d]} \frac{1}{|\mathcal{S}_{[t],k}^{(m)}|} |\epsilon^\top X^{(r)}| \geq \lambda_0(\chi, \frac{\phi\sqrt{p}}{2}) \right)$$
$$+ \mathbb{P}\left( \hat{\Sigma}((\mathcal{S}_{[t],k}^{(m)})') \notin \mathcal{C}(\mathrm{supp}(\beta), \frac{\phi}{\sqrt{2}}) \right)$$
$$+ \mathbb{P}\left( \frac{|(\mathcal{S}_{[t],k}^{(m)})'|}{|\mathcal{S}_{[t],k}^{(m)}|} \leq \frac{p}{2} \right) \tag{37a}$$
$$\leq 2\frac{1}{Mt} + \exp\left( -\frac{M^2 p_*^2 C_2(\phi)^2}{16} t \right) + \exp\left( -\frac{M^2 p_*^2}{36} t \right)$$
$$\leq \frac{2}{Mt} + 2\exp\left[ -\frac{p_*^2 C_2(\phi_0)^2}{32} \cdot M^2 t \right]$$

$\square$

# D Proof of Theorem 3.7

We break the proof into 3 steps.

**Step 1. Instantaneous regret.** Given Lemma 3.6, if $t > C_5$, we have

$$r_{t+1}$$

$$\leq 2bx_{\max}\mathbb{E}\left[\sum_{i\in\hat{\mathcal{K}}}\mathbb{I}\left[\left(X_{t+1}^\top\hat{\beta}_i \geq X_{t+1}^\top\hat{\beta}_1\right)\cap B_i\right]\right] + 2\delta x_{\max}\mathbb{E}\left[\sum_{i\in\hat{\mathcal{K}}}\mathbb{I}\left(B_i^c\right)\right]$$

$$\leq 2bx_{\max}\Pr\left[\left\|\beta_1 - \hat{\beta}_1\right\|_1 > \delta\right] + \Pr\left[\left\|\hat{\beta}_i - \beta_i\right\|_1 > \delta\right] + 2\delta x_{\max}2C_0\delta x_{\max}$$

$$= 4bx_{\max}(\frac{2}{Mt} + 2\exp\left[-\frac{p_*^2 C_2\left(\phi_0\right)^2}{32}\cdot M^2 t\right]) + 4C_0 x_{\max}^2\delta^2$$

$$= 8bx_{\max}(\frac{1}{Mt} + \exp\left[-\frac{p_*^2 C_2\left(\phi_0\right)^2}{32}\cdot M^2 t\right]) + 1024\frac{C_0 x_{\max}^2}{p_*^3 C_1}\frac{\log t + \log M + \log d}{Mt}$$

$$\leq 8bx_{\max}(1 - \frac{32}{p_*^2 C_2\left(\phi_0\right)^2})\frac{1}{M^2 t} + 1024\frac{C_0 x_{\max}^2}{p_*^3 C_1}\frac{\log t + \log M + \log d}{Mt}$$

$$= L_1\frac{1}{M^2 t} + L_2\frac{\log t + \log M + \log d}{Mt}$$

The last inequality is due to $\exp(-x) < 1/x$ for all $x > 0$. The constants are $L_1 = 8bx_{\max}(1 - \frac{32}{p_*^2 C_2(\phi_0)^2})$ and $L_2 = 1024\frac{C_0 x_{\max}^2}{p_*^3 C_1}$.

**Step 2. Cumulative Regret of one client.**

$$\sum_{t=C_5}^T r_{t+1} \tag{38}$$

$$= \frac{1}{M}\left[\frac{L_1}{M}\sum_{t=C_5}^T\frac{1}{t} + L_2\sum_{t=C_5}^T\frac{\log t + \log M + \log d}{t}\right] \tag{39}$$

$$= \frac{1}{M}\left[\frac{L_1}{M}\log T + L_2\left((\log T)^2 + \log(Md)\log T\right)\right] \tag{40}$$

$$= \frac{1}{M}\left[\frac{L_1}{M}\log T + L_2\log(Md)\log T + L_2(\log T)^2\right] \tag{41}$$

**Step 3. Cumulative regret of all client**

$$R_T = \sum_{m=1}^M\sum_{t=1}^T r_t^{(m)}$$

$$= \sum_{m=1}^M\left(\frac{C_5}{M}2bx_{\max} + \frac{1}{M}\left[\frac{L_1}{M}\log T + L_2\log(Md)\log T + L_2(\log T)^2\right]\right)$$

$$= C_5 2bx_{\max} + \left(\frac{L_1}{M} + L_2\log(Md)\right)\log T + L_2(\log T)^2$$

$$= O(\log M(\log T)^2)$$

## E  Supporting Lemmas

Here we state a general version of Lemma 1 in Bastani & Bayati (2020) to facilitate our analysis.

**Lemma E.1.** *(Lasso deviation inequality for non i.i.d. dataset) Given a dataset $\mathcal{D}$ with its i.i.d. subdataset $\mathcal{D}'$. Let $\hat{\beta}(\mathcal{D}, \lambda)$ be the Lasso regression estimator (Definition 1.1) trained on $\mathcal{D}$ with regularization level $\lambda$. Suppose the population covariance matrix $\Sigma$ satisfy the compatability condition $\mathcal{C}(supp(\beta), \phi)$. Let $\chi > 0$ be an error upper bound specified by user. Set $\lambda\left(\chi, \phi\sqrt{p}/2\right) = \chi\phi^2 p/(16 s_0)$. Then, the $l_1$ estimation error of $\hat{\beta}(\mathcal{D}, \lambda)$ satisfies the following deviation inequality:*

$$\mathbb{P}\big(\|\hat{\beta}(\mathcal{D}, \lambda) - \beta\|_1 \geq \chi\big) \leq \alpha(\chi), \tag{42}$$

*where*

$$\alpha(\chi) = \mathbb{P}\left(\max_{r \in [d]} \frac{1}{|\mathcal{D}|}|\epsilon^\top X^{(r)}| \geq \lambda_0(\chi, \frac{\phi\sqrt{p}}{2})\right) + \mathbb{P}\left(\hat{\Sigma}(\mathcal{D}') \notin \mathcal{C}(supp(\beta), \frac{\phi}{\sqrt{2}})\right) + \mathbb{P}\left(\frac{|\mathcal{D}'|}{|\mathcal{D}|} \leq \frac{p}{2}\right) \tag{43}$$

## F  Experiment Details and Additional Experiments

### F.1  Synthetic Data

For the baseline case, we set $d = 100$, $K = 5$, $M = 5$ and $s = 5$, where $s$ is the sparsity level of the parameters that should satisfy $s \ll d$. The $\beta$ of each arm for each client is generated by first choosing the possibly nonzero positions for all clients. In our case we randomly choose 10 positions for each arm and the 10 positions are shared across the different clients to represent that the bandit environment faced by different clients are similar (but not exactly the same). We then choose the $s$ non-zero elements randomly within the 10 positions of each arm for each client, and then generate the values from a uniform distribution on $[0, 1]$. Note that the nonzero values of each arm across different clients are different, and the positions are different too. The covariates of each client are generated from a standard normal distributions independently.

We provide some additional experimental results for different choices of $d, K, M$ in Figure 5 for readers interested in the effect of changing these hyper-parameters in our synthetic data.

### F.2  PharmGKB

We utilize the publicly available code for PharmGKB at https://github.com/chuchro3/Warfarin for the processing of the dataset. Specifically, the covariates are generated from either the value in the dataset if it is numerical or from a bag-of-words of all the categories if categorical. Then the covariates are transformed into one-hot encodings and thus the vector is very high-dimensional ($d = 5528$). We follow Bastani & Bayati (2020) and classify the different dosages of Warfarin into three classes. $[0, 3]$ is the low-dosage class, $[3, 7]$ is the moderate dosage class, and $[7, \infty]$ is the high-dosage class. The reward is defined to be 1 if the correct dosage class is chosen and 0 if not, and we add standard Gaussian noise to the reward. The error rate (in Figure 3) is defined by the number of wrong classifications divided by the total number of classifications.

### F.3  Medical MNIST

**Neural Network Setting**. The fully-connected neural network used to train on the Medical MNIST dataset has the architecture shown in Table 2, where *input size* is the size of the vectorized image, which is 10800 in our case and *output size* is the number of classes, which is 6. The dataset was randomly split into a training set and a testing set in the ratio of 9:1. The images are pre-processed with random rotation, random horizontal flipping, resizing, center cropping and normalization. The training batch size was 32. The optimizer was SGD with learning rate 0.01 and no momentum or weight decay. The neural network was trained on the training set for 10 epochs and evaluated on the testing set. The final testing classification accuracy was around 99%.

**Bandit Setting**. The number of arms for each agent is equal to the number of classes (6) in this problem. We use the output of the last ReLU layer (i.e., before the last layer) as the feature vector for bandit problems,

which means that the feature dimension is 500. At each round, The regret is defined to be whether the feature vector is correctly classified, i.e., the reward is 0 if the the chosen arm is different from the class of the image and 1 if they are the same. Standard Gaussian noise is added to the reward received by the bandit algorithms.

Table 2: Architecture of the neural network

| |
|---|
| *inLayer*: Linear Layer(*input size*, 4000) |
| ReLU() |
| *hidden1*: Linear Layer(4000, 2000) |
| ReLU() |
| *hidden2*: Linear Layer(2000, 1000) |
| ReLU() |
| *hidden3*: Linear Layer(1000, 500) |
| ReLU() |
| *outLayer*: Linear Layer(500, *output size*) |

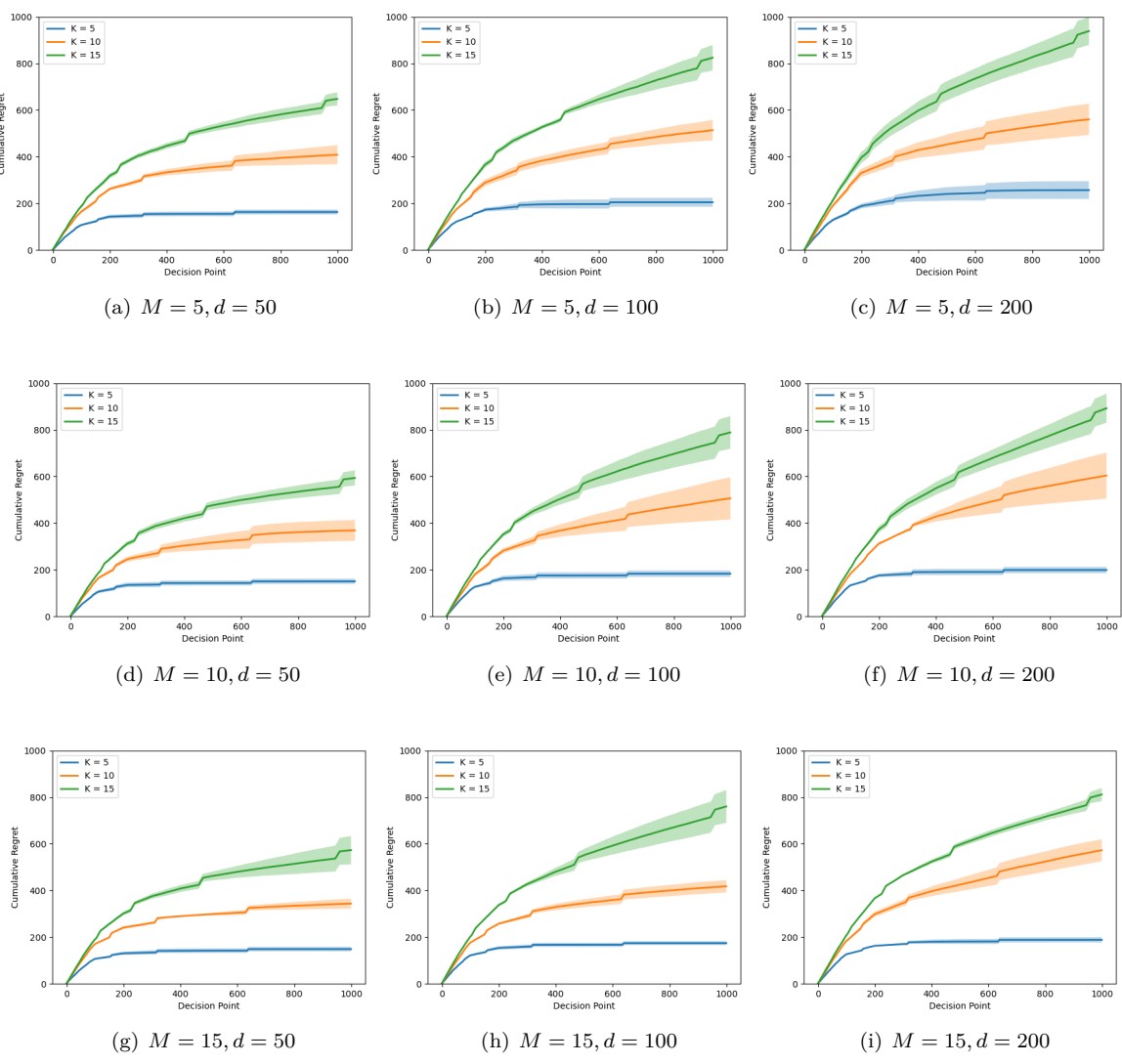

Figure 5: Comparison of the the average cumulative regret per client of the Fedego `Fedego Lasso` algorithm under different settings of the parameters $M, K, d$ on the synthetic dataset

