# OpenReview forum: "Federated High-Dimensional Online Decision Making"
_TMLR — Accepted by TMLR_

### Review · Reviewer_MhHZ · 2023-02-12

**Summary Of Contributions:**

This paper studies federated lasso bandits with $M$ clients that can both play the linear bandit game in parallel and collaborating through a server to aggretate the lasso model. The design objective is to minimize the cumulative regret while also protecting local information leakage for each client. The proposed algorithm interlaces collaboration phases where the agents work together with individual decision making phases where client only tries to maximize her own reward. The data from the former phase (collaboration) is used to train local lasso models that will be periodically transmitted to the server for aggregation, which will then be sent back to the clients. The algorithm is analyzed and evaluated empirically.

**Audience:**

Yes

**Broader Impact Concerns:**

No such need identified.

**Claims And Evidence:**

No

**Requested Changes:**

The paper must be substantially re-written to clarify the key points. The four concerns in the previous box should all be addressed.

**Strengths And Weaknesses:**

This work has some interesting ideas and is looking at a potentially important problem. However, there are several critical issues associated with the current form of the paper.

(1) Clarity. The paper is very difficult to follow. The figures have a lot of jumping back and forth. The algorithm explanations following each of Alg. 1, 2, 3, and 4 are terrible and do not make much sense. It almost seems that the paper is cut to the current page limitation without thinking about the coherence. As a result, it is very difficult to evaluate.

(2) Communication cost and privacy are not well investigated. The authors made a big deal (repeatedly) about the privacy aspects of the prior works, but failed to present any formal privacy advantage of the proposed solution. The only thing is that communicating model instead of data is "more private", which is far from sufficient. As for communication cost, there is always a tradeoff between more communication and better performance, and this aspect is entirely missing.

(3) Lasso and high dimensionality. It is unclear why the lasso regression model is particularly useful for this problem. The authors claimed that this is due to high dimensionality. However, all the analysis does not really use high dimensionality. It is very generic.

(4) Novelty. If $M=1$, does it go back to the known result? It seems the only new thing is to add "federated lasso model", i.e., sending local lasso models to server for aggregation, and then sending it back to clients.

---

> ### Author Response · Authors · 2023-04-07
> **Thank you for your comments!**
>
> Thank you for your valuable comments! We want to address your concerns one by one.
>
> ***(1) Clarity. The paper is very difficult to follow. ... As a result, it is very difficult to evaluate.***
>
> Thank you for bringing up these unclear sections! We will follow your suggestions and add more detailed descriptions of the algorithms to Section 3 in the revised version and reorganize the content to make it clearer.
>
>
> ***(2) Communication cost and privacy are not well investigated. The authors made a big deal (repeatedly) about the privacy aspects of the prior works, but failed to present any **formal privacy advantage of the proposed solution**..., and this aspect is entirely missing.***
>
>
> The privacy guarantee in our paper is provided through the limited number of communication rounds in our algorithm. As shown in Table 1 and Section 2.1C, our communication cost is logarithmic with respect to the number of rounds $T$, which indicates that little information is shared among the clients and thus privacy is protected. Such privacy guarantee is standard in federated bandit problems and commonly observed in related works such as [1], [2], and [3]. For the tradeoff, we can decrease $q$ to increase communication frequency, as long as $q$ is greater than the $q_{\textit{teamwork}}$ defined at equation (3), Section 3.2.
>
>
> [1]. Chengshuai Shi, Cong Shen, and Jing Yang. Federated multi-armed bandits with personalization. In International Conference on Artificial Intelligence and Statistics, pp. 2917–2925. PMLR, 2021
>
> [2]. Ruiquan Huang, Weiqiang Wu, Jing Yang, and Cong Shen. Federated linear contextual bandits. In Thirty Fifth Conference on Neural Information Processing Systems, 2021.
>
> [3]. Wenjie Li, Qifan Song, Jean Honorio, Guang Lin. Federated X-armed bandit. arXiv 2205.15268. 2022.
>
>
>
>
> ***(3) Lasso and high dimensionality. It is unclear why the lasso regression model is particularly useful for this problem... It is very generic.***
>
>
> **Why Lasso is useful?** We use Lasso since our application is in e-commerce and medical setting. In such setting, the context vector is often high-dimensional, but only few number of variable are decisive. Lasso captures such property by assuming the sparse structure of the regression coefficients.
>
> **All the analysis does not really use high dimensionality.**
> The reason of not really use high-dimensionality is due to the sparsity of regression coefficient $\left\|\beta_k\right\|_0 \leq s_0$. With such structure of regression coefficient, we leverage the technique in high-dimensional statistic in our theory.
>
> The Lasso bandit article [4] explains that in medical applications, we might want to identify a small number of biomarkers that are associated with a certain disease. In e-commerce applications, we might want to select a small number of features that are relevant for predicting user behavior. Both of these problems can be formulated as a sparse linear regression problem, where we want to minimize the number of non-zero coefficients in the regression model.
>
> Lasso is particularly useful in these cases because it performs both feature selection and regularization simultaneously. By penalizing the L1 norm of the regression coefficients, Lasso encourages sparsity and tends to produce models with a small number of non-zero coefficients. This can lead to better generalization performance and easier interpretability of the resulting model.
>
> Therefore, the motivation for using Lasso is to take advantage of its ability to perform feature selection and regularization simultaneously, which is particularly useful for applications where sparsity is assumed.
>
>
> [4]. Hamsa Bastani and Mohsen Bayati. Online decision making with high-dimensional covariates. Operations
> Research, 68(1):276–294, 2020

---

### Review · Reviewer_EENi · 2023-03-07

**Summary Of Contributions:**

The paper introduces Fedego Lasso, a novel algorithm for the Federated bandits setting that offers sub-linear theoretical regret and preserves local client privacy to safeguard sensitive decision-making information. The proposed approach is supported by both theoretical and empirical analyses, demonstrating its effectiveness in achieving the stated objectives.

**Audience:**

Yes

**Broader Impact Concerns:**

As noted in the weakness section, it is essential to provide a more detailed explanation of the local client privacy argument, including a clear definition of what constitutes local privacy "information" and why sharing it could be potentially harmful. This clarification is particularly important for readers outside the federated learning community, who may not be familiar with the underlying concepts and implications. By providing a comprehensive explanation, the paper can better contribute to the broader scientific community's understanding of this critical topic.

**Claims And Evidence:**

Yes

**Requested Changes:**

It would be beneficial to emphasize the Lasso estimation technique in the algorithm box to facilitate the reader's understanding without the need to search for the definition of different $\beta$ in the paper. This would help readers quickly grasp the essence of the algorithm and reduce the cognitive load required to understand it.

**Strengths And Weaknesses:**

### Strengths:
-   The paper presents rigorous mathematical assumptions and proofs, demonstrating a sound and solid approach to the problem.
-   The theoretical and empirical results are well-founded and support the effectiveness of the proposed algorithm.
-   The high-level design of the algorithm is well-explained and easy to understand through the demonstration in Figures 1 and 2.

### Weaknesses:
-   The primary motivation behind the paper is not entirely clear. Although the paper argues that existing works on federated bandits only protect the local privacy of data but not the local privacy of "information," the definition of local privacy of "information" is not well-defined. Therefore, further elaboration on this motivation would be beneficial.
-   The paper only compares its proposed algorithm with Lasso Bandits in the experimental setting, and other baseline algorithms are not included. It would be useful to add additional baseline algorithms or provide explicit reasoning for the exclusion of other baselines (e.g., different privacy settings).

---

### Review · Reviewer_yVoM · 2023-05-04

**Summary Of Contributions:**

This paper proposes a new federated bandit policy design named Fedego Lasso for high-dimensional online decision-making. The new policy achieves sublinear regret and preserves the privacy of local clients by only sharing parameter estimates instead of the raw data. The paper provides theoretical guarantees and empirical experiments on synthetic and real datasets.

**Audience:**

Yes

**Claims And Evidence:**

Yes

**Requested Changes:**

1. A more extensive analysis of how the new algorithm improves compared to Teamwork Lasso and Lasso Bandit.

2. The paper claims that the "exploration" process would benefit from all local clients exploring the same arm. Though I see the intuition behind this, will it be a good idea if a proportion of clients explore the same arms?

3. Can we compare with Teamwork Lasso in Figure 3?



**Strengths And Weaknesses:**

**Strengths**

1. The paper is well-written, with concise descriptions and theoretical analysis. Both the "Teamwork" and "Selfish" stages are clear to me.

2. The paper conducts extensive experiments on both synthetic and real datasets to show the effectiveness of the proposed algorithm. The comparison with existing algorithms in Table 1 is very helpful.

**Weakness**

1. While I do see the motivation and effects of the new algorithm design, I suggest a more extensive analysis of how the new algorithm improves compared to Teamwork Lasso and Lasso Bandit. Now there is a paragraph saying the Teamwork Lasso Bandit has privacy leakage by sharing the raw data, but it would be beneficial to analyze the influences of not touching the raw data more thoroughly (e.g., how the performance is influenced?). Also, Figure 3 only compares with Naive Lasso, not the Teamwork Lasso.

2. The paper primarily focuses on linear contextual bandits in federated settings. I wonder whether the approach can extend to other bandit settings. What are the challenges of such extension?

---

> ### Author Response · Authors · 2023-05-18
> **Reply to Reviewer yVoM**
>
> Thank you for your valuable comments! We hope to address your concerns as follows.
>
> **Regarding Weakness 1**
>
> We will emphasize the improvements of our algorithm in terms of practical applicability to Teamwork Lasso and Lasso Bandit. Lasso Bandit assumes that the data is only used by personnel within the institution, so its policy design assumes that the Bandit algorithm can collect and analyze data according to its own preferences. However, in the scope of our paper, multiple institutions (clients) shares data (or information) for joint decision-making. If each institution agrees to share their data without any restrictions or privacy regulation, then the results of Teamwork Lasso can address the decision effectiveness in this "complete data sharing" scenario. However, in reality, each executing institution (client) is unwilling and legally not allowed (e.g., in the healthcare field with HIPPA regulations) to freely exchange data. This forces each institution to only share the information they are comfortable in their own data (e.g., the results of Lasso Regression), which is the setting considered in our article on Fedego Lasso. We anticipate that not accessing the original data may affect the efficiency of decision-making (we will conduct experiments to investigate this further in the revision), but once the sample size is sufficient, we can still achieve decision-making effectiveness similar to  "complete data sharing" scenario. We will include a comparison with Teamwork Lasso in the revision.
>
> **Regarding Weakness 2**
>
> We believe that our policy design can be extended to more sophisticated reward models (e.g., generalized linear models). In Appendix B, the proof of Lemma 3.3 can be generalized to more refined reward models as long as we have a good understanding of the statistical estimation's tail estimate for that model. In principle, a format result on the statistical estimation's tail estimate will  enjoy a direct extension of the similar regret analysis in our work. However, if we apply Bandit to a more practical level, one aspect to consider is how to increase the fairness of decision-making in a given setting. As you mentioned in Requested Changes 2, if a subset of customers keeps exploring the same arms, it is challenging to ensure fairness in the allocation of experimental resources. These customers may miss the opportunity to try their best arm, bearing the experimental costs for joint research. To address this issue, a significant challenge we anticipate is to "define a regret formulation that sufficiently reflects decision fairness." Are there minority groups among the clients? How can we ensure that the results of joint decision-making benefit everyone instead of some clients bearing the costs while others enjoy the outcomes? These are non-trivial questions and challenges we expect to encounter.
>
> Thank you for your suggestions! We will incorporate all the requested changes in the revision. We will appreciate it if you could review our revision when it's ready.

---

### Author Response · Authors · 2023-05-29
**New Version**

Dear Reviewers, Action Editor, and Editors in Chief,

We would like to thank you again for spending your precious time in reviewing our paper and providing suggestive comments. We have read and replied to all your reviews. To address your concerns, a new version of our paper is uploaded with the changes highlighted in red. Specifically,

* Before section 1.1, we have added a more extensive explanation of how the new algorithm improves over Teamwork Lasso and Lasso Bandit.

* In Remark 1.3, we have provided more intuition on why  the lasso regression model is particularly useful for this problem.

* After Lemma 3.1, we have added more explanations on why the exploration process would benefit from all local clients exploring the same arm.

* We have refined the notations in Algorithm 1, 2, 3 & 4 and added more algorithm explanations to make our algorithm easier to understand.

* For all the experiments, we have added Teamwork LASSO (Wang et al, 2020) as a baseline.

Please let us know if you think anything else needs to be changed. Thank you!

Best Regards,

Paper 761 Authors

---

### Decision · Action_Editors · 2023-06-24

**Recommendation:** Accept with minor revision

**Comment:**

The reviewers are all leaning toward accepting the paper.

I have a number of comments --- in addition to the reviewers' comments that were addressed by the authors in a revision.

1. As a paper with new technical results, it is missing a technical discussion on what is really new / different in the analysis.  While TMLR's criteria of acceptance does not include novelty, it is still required to properly discuss what is borrowed from related work and what is the "delta". What is new in the regret analysis over Bastani & Bayati (2020) now that you operate in the federated learning case.

2.The discussion w.t.t. (Kim & Paik, 2019; Hao et al., 2020; Oh et al., 2021; Li et al., 2021) is unsatisfactory. While these work is about linear bandits, not contextual linear bandits, one can reduce the problem from this paper to the linear bandits problem by concatenating the  \beta_k into a longer parameter vector, and creating "one-hot" features for each arm. I think a theoretical comparison with respect to these work should be there.   The current discussion sounds like these lasso-bandit-lie methods

3. The paper emphasizes quite a bit on "privacy protection", "local privacy " and so on, but the definition of "privacy" is not stated clearly.

4. The presentation and clarity should be improved. For example , ”under data decentralization with a local privacy protection argument.“  What is a "local privacy protection argument"?

I hope the authors could take the chance of a "minor revision" and address the above issues. I believe it will make the paper more impactful.

**Audience:**

The paper would be of interest to researchers interested in either contextual bandits / linear bandits who are also interested in federated learning.

**Claims And Evidence:**

The main technical claim is an algorithm for federated contextual linear bandits problem with polylog(d,T)  regret bound --- thus handling the high-dimensional (large d) case. Proof is included and reviewers are confident about the correctness of the technical result.

The experiments are good.

---

> ### Author Response · Authors · 2023-07-10
> **Thank you and new version uploaded**
>
> Dear Action Editor,
>
>   We would like to thank you for accepting our paper providing these useful and thorough final reviews. Sorry for this late response.
>
>   We have uploaded a new version (pre-camera-ready) of our paper, where we hopefully have addressed all your comments and questions. The changes from the previous version are highlighted in red. Specifically, we have changed the following
>
>   1. For Q1 in your review, we have provided more discussion on what is new in our paper and the analysis in page 3 to highlight our contributions.
>
>   2. For Q2, we have added more discussions in page 3-4 and page 11 to compare our work with the other line of research (Kim and Paik, 2019, Hao et al, 2020, Oh et al, 2021).  Our work is indeed in the line of LASSO Bandit (Bastani and Bayati, 2020).
>
>   3. For Q3, we have provided more information on how our work protects user privacy in page 5.
>
>   4. For Q4, we have updated page 1 to  discuss the meaning of "local privacy protection"
>
>   We hope that you are satisfied with this revision and our paper can be accepted. Please let us know if you think anything else needs to be changed.
>
>   Again, thank you so much for your help!
>
> Sincerely,
> Authors

---

> ### Author Response · Authors · 2023-08-15
> **Camera-ready Version Uploaded**
>
> Dear Action Editor and Reviewer,
>
>   Since we have not yet received any additional change requests and comments in the past month, we have uploaded the current version as the camera-ready version. Please let us know if you believe any problems/issues are still not addressed.
>
>   Thank you so much!
>
> Sincerely,
> Authors